# Updates of Risk Factors for Anastomotic Leakage after Colorectal Surgery

**DOI:** 10.3390/diagnostics11122382

**Published:** 2021-12-17

**Authors:** Eugenia Claudia Zarnescu, Narcis Octavian Zarnescu, Radu Costea

**Affiliations:** 1Department of General Surgery, “Carol Davila” University of Medicine and Pharmacy, 050474 Bucharest, Romania; eugenia.zarnescu@umfcd.ro (E.C.Z.); radu.costea@umfcd.ro (R.C.); 2Second Department of Surgery, University Emergency Hospital Bucharest, 050098 Bucharest, Romania

**Keywords:** anastomotic leakage, anastomotic fistula, risk factors, colorectal surgery, colorectal cancer

## Abstract

Anastomotic leakage is a potentially severe complication occurring after colorectal surgery and can lead to increased morbidity and mortality, permanent stoma formation, and cancer recurrence. Multiple risk factors for anastomotic leak have been identified, and these can allow for better prevention and an earlier diagnosis of this significant complication. There are nonmodifiable factors such as male gender, comorbidities and distance of tumor from anal verge, and modifiable risk factors, including smoking and alcohol consumption, obesity, preoperative radiotherapy and preoperative use of steroids or non-steroidal anti-inflammatory drugs. Perioperative blood transfusion was shown to be an important risk factor for anastomotic failure. Recent studies on the laparoscopic approach in colorectal surgery found no statistical difference in anastomotic leakage rate compared with open surgery. A diverting stoma at the time of primary surgery does not appear to reduce the leak rate but may reduce its clinical consequences and the need for additional surgery if anastomotic leakage does occur. It is still debatable if preoperative bowel preparation should be used, especially for left colon and rectal resections, but studies have shown similar incidence of postoperative leak rate.

## 1. Introduction

Anastomotic leak (AL) after colorectal surgery is a major complication, increasing postoperative morbidity and mortality. The incidence rate of anastomotic leak after colorectal surgery was reported to be between 2% and 19% and the mortality related fistula was reported to be between 0.8% and 27% [1,2,3,4]. Anastomotic leakage has a significant negative impact on disease-free survival, overall survival and local recurrence [5,6]. A meta-analysis including a total of 154,981 patients showed that anastomotic leakage had a negative impact on overall survival [7].

Differences between studies regarding anastomotic leakage rate result from heterogeneity of anastomotic fistula definitions. Different AL rates were reported if the fistula was diagnosed clinically, radiologically, endoscopically or intraoperatively [8,9,10,11]. The International Study Group of Rectal Cancer published specific guidelines about the definition of anastomotic leak and a grading system of severity [12]. Later, multiple studies were published that modified the Delphi consensus on the definition and management of anastomotic leakage in colorectal surgery [13,14,15].

There are multiple studies in the literature that have identified numerous risk factors (Table 1) associated with anastomotic fistula after colorectal surgery, factors that can be divided into local and general factors; pre-, intra- or postoperative factors; and modifiable or non-modifiable factors [16,17]. Identification of risk factors for AL can help surgeons in clinical practice to use a tailored approach for decision making. Multiple studies have identified several risk factors, but unfortunately it is not still possible to perfectly predict the occurrence of fistula for a specific patient. Pre- or intra-operative decisions about whether to perform an anastomosis or a stoma remain difficult. Several leakage scores were developed to help surgeons to provide an objective assessment of the risk of AL and in making a decision about surgical management [18,19,20]. The anastomotic leakage requires further evaluation of its grade of severity that will decide the ultimate management strategies.

## 2. Preoperative Risk Factors

**Male gender** has been shown to be an independent risk factor for leakage in all types of colorectal anastomosis [21,22,23]. Jannasch et al. found in their study that leakage was 1.7 times more frequent in men [21]. Anastomosis in the narrower male pelvis results in more difficult resection for men, in both open and laparoscopic colorectal surgery [24]. In a recent study of 429 patients with rectal resections and colorectal anastomosis, the authors found male gender (OR 3.8; 95% CI 1.9−7.7; *p* < 0.001) to be an independent variable associated with increased AL rate [25]. The influence of androgen-related differences in the intestinal microcirculation may be involved in anastomoses healing [26]. An experimental study on rats showed a less favorable collagen metabolism in colonic anastomoses of males compared with females during early wound healing [27].

Use of **alcohol and smoking** is known to have a negative effect on general wound healing [28]. Smoking history and current smokers have a significantly increased risk for leakage. The relationship between the two might be secondary to ischemia caused by smoking-related microvascular disease [29]. Kwak et al. reported habitual smoking to be significantly associated with AL (OR 6.529, *p* = 0.007), and it was suggested that vascular ischemia from nicotine-induced vasoconstriction and microthromboses, together with carbon monoxide-induced cellular hypoxia, inhibits anastomotic circulation in smokers [30]. Alcohol history was also associated with a higher risk of anastomotic leak in several publications [21,29]. Large quantities of alcohol consumption might be a surrogate for poor nutritional status. 

The **American Society of Anesthesiologists** (ASA) score has been shown to be a significant risk factor for postoperative fistula in some studies [31,32]. An ASA grade equal or greater than 3 was reported to be an independent risk factor for anastomotic leakage [16,21,24,33]. The presence of comorbid conditions in patients who underwent colorectal surgery was shown to be a risk factor for leakage. **Diabetes mellitus** [34,35,36], **cardiovascular disease** [37], **obstructive pulmonary disease** [36] and **renal failure** [38] resulting in a higher **Charlson**
**Comorbidity Index (CCI)** score [23,32,39] were reported as risk factors. Tian et al. found that patients with a CCI score ≥ 3 had 1.82 times higher risk of anastomotic leakage compared with patients with a CCI score of zero [39].

**Weight and nutrition status** are important factors during the evaluation of patients with colorectal anastomosis. Weight loss and malnutrition before surgery have an important role in anastomotic dehiscence, some studies supporting this association [3]. Usually, malnutrition is associated with other factors influencing the healing process [40]. Kwag et al. in their analysis concluded that only patients at nutritional risk have higher complication rates after colorectal surgery [41]. There is evidence in the literature to suggest that obesity becomes a risk factor for leaks in very low rectal anastomoses because it may be related to tension at the anastomotic site. A body mass index higher than 30kg/m2 has been shown to be an independent factor for anastomotic leak in some studies [17,42,43]. Some authors found that the measurement of visceral fat on CT scan examination is a more sensitive factor than body mass index (BMI) to predict development of anastomotic dehiscence [44,45]. Goulart et al. showed a direct relationship between visceral fat and anastomotic leakage and reoperation [46]. A meta-analysis evaluating visceral fat in patients with laparoscopic colorectal surgery revealed that visceral obesity was associated with longer operative time, less lymph nodes harvested, more conversion to open procedure, higher morbidity, more surgical site infection and more anastomotic leakage [47].

There were authors reporting the **preoperative albumin level** less than 3.5 g/dL as being a significant factor for leakage [33,48,49]. In a recent study, the authors found no significant difference in preoperative serum albumin level between the anastomotic leakage group and the non-anastomotic leakage group, but the postoperative serum albumin level was significantly lower in the anastomotic leakage group [50]. In this study, a lower level of serum albumin (less than 3.2 g/dL) on postoperative days 1 and 3, a higher count of leukocytes on postoperative days 1 and 3, and surgery for rectal cancer were independent risk factors for anastomotic leakage.

**Nonsteroidal anti-inflammatory drugs (NSAIDs**), which are commonly used analgesic and anti-inflammatory adjuncts, have many physiologic effects and are being used more commonly to treat postoperative pain, but recent small studies have suggested that NSAIDs may impair anastomotic healing [51,52,53]. In a study by Gorissen et al., patients on NSAIDs had higher anastomotic leakage rates than those not treated with NSAIDs (13.2% versus 7.6%; *p* = 0.010) [54]. However, a multicenter retrospective study found no statistically significant increase in the proportion of patients with anastomotic leak when prescribing nonsteroidal anti-inflammatory drugs for analgesia in the early postoperative period for patients undergoing elective colorectal surgery [55]. This finding is supported by other studies, showing that use of NSAIDs did not increase the risk of anastomotic leakage after anterior resection for rectal cancer [56,57]. Prolonged use of **corticosteroids** was proposed as a risk factor for anastomotic leakage [58,59,60]. A prospective study found a significantly increased incidence of anastomotic dehiscence in patients treated with long-term corticosteroids and perioperative corticosteroids for pulmonary comorbidity [61].

**Mechanical bowel preparation** (MBP) in colorectal surgery has been used for decades, despite increasing evidence challenging its benefits [62,63]. The reason for using MBP is that it reduces fecal bulk, clears the bowel lumen and therefore reduces bacterial colonization, thus decreasing the risk of postoperative complications such as anastomotic dehiscence and wound infection [64,65]. On the other hand, MBP has its own complications, such as clinically significant dehydration and electrolyte disturbances in the preoperative period [66,67], and the process is both time-consuming and unpleasant for patients [68]. Opponents of this practice sustain that use of oral and intravenous prophylactic antibiotics are sufficient because the evidence has shown that the gut microbial flora load is not reduced grossly by bowel preparation [69].

In a meta-analysis by Rollins et al. including 21,568 patients undergoing elective colorectal surgery, the authors concluded that the use of MBP versus either absolutely no bowel preparation or a single rectal enema was not associated with a statistically significant difference in the incidence of anastomotic leak, surgical site infection, intra-abdominal collection, mortality, reoperation, or total length of hospital stay [70]. This evidence was supported by other studies [64,71]. Several studies focusing on rectal surgery suggested that mechanical bowel preparation could be used selectively, even though no significant effect was found [72,73]. 

In recent years, authors have shown that a combined preoperative mechanical and oral antibiotics bowel preparation resulted in a significantly decreased risk of overall morbidity, superficial surgical site infection, anastomosis leakage and intra-abdominal infections when compared to no preoperative bowel preparation [64,74]. Comparative results assessing the impact of mechanical bowel preparation, with or without oral antibiotics, on postoperative anastomotic leakage are described in Table 2. A retrospective study on 40,446 patients concluded that a combined regimen of oral antibiotics and mechanical bowel preparation offered no superiority when compared with oral antibiotics alone in terms of surgical site infection, anastomotic leak, postoperative ileus and major morbidity after elective colorectal surgery [75]. In conclusion, there is a lack of consensus regarding the use of mechanical bowel preparation due to inconsistent results of the incidence of postoperative complications, including the anastomotic leakage.

**Preoperative chemoradiotherapy** is part of multimodal treatment and is generally recommended for patients with locally advanced rectal cancer followed by TME surgery. It is accepted that these therapeutic modalities can reduce the local recurrence rate [77,78]. There are some retrospective studies that have reported an association between preoperative radiotherapy and anastomotic leak [22,79]. A randomized controlled trial on 318 patients with rectal cancer concluded that preoperative radiotherapy increases the risk of anastomotic leakage. The anastomotic leak rate was 20.2% in patients receiving preoperative radiation and 5-fluorouracil alone and 23.6% if this therapy was combined with oxaliplatin comparing with 8.5% in patients with preoperative chemotherapy without radiation (*p* = 0.007) [80]. Prospective trials and cohort studies have shown no statistically significant association between neoadjuvant treatment and anastomotic leakage. A Dutch trial comparing TME plus preoperative radiotherapy versus TME alone reported that there was no significant difference in anastomotic leak rates [81]. In a report using propensity score matching analysis, Chang et al. showed that in patients with rectal cancer, preoperative chemoradiotherapy did not increase the risk of postoperative anastomotic leak after low anterior resection [82]. These results were sustained by other authors in a meta-analysis, showing that neoadjuvant therapy does not appear to increase the incidence of postoperative anastomotic leakage after anterior resection for rectal cancer [83,84]. The meta-analysis of Hu et al. indicated that the incidence of AL was not significantly increased after short-course preoperative RT (OR = 1.19 [95% CI: 0.89–1.60; *p* = 0.25). There was no increase of AL (OR = 1.38; 95% CI: 0.75–2.54; *p* = 0.31) in patients who had been treated with long-course of preoperative RT [83]. Meta-analysis of Ma et al. showed that preoperative RT (PRT) and preoperative chemoradiotherapy (PCRT) significantly increased the incidence of wound problems (PRT: OR = 1.43, 95% CI = 1.17–1.74, *p* < 0.01; PCRT: OR = 1.52, 95% CI = 1.08–2.16, *p* = 0.02), but not the incidence of anastomotic leakage or bowel obstruction [84]. In this study, the short course and the long course preoperative radiotherapy had similar rates of anastomotic leakage. In addition, the interval to surgery after neoadjuvant therapy and preoperative radiotherapy was not associated with an increased incidence of postoperative leak [37,83,85]. Several studies (Table 3) comparing the impact of neoadjuvant therapy on anastomotic leakage are provided for comparison.

## 3. Intraoperative Risk Factors

One of the most important risk factors for anastomotic leak is the **distance of the suture from the anal verge**. Zhang W. et al. showed in a study of 319 patients with middle and low rectal cancer resection that a distance of anastomosis less than 7 cm from the anal verge is an independent risk factor for leakage [34]. Most studies defined a low rectal anastomosis as an anastomosis 5 cm or less from the anal verge. Rullier et al. have shown a leak rate 6.5 times higher in anastomoses located less than 5 cm from the anal verge [87]. In another study of 1392 patients with colorectal cancer, the anastomotic leak rate was 4.7% in case of extraperitoneal anastomosis compared with 0.2% in intraperitoneal anastomosis [88]. A meta-analysis on the six studies involving rectal resections only found that a low rectal anastomosis was associated with a high risk of leakage [89]. On the other hand, a retrospective cohort study of 9192 patients with colorectal resections showed no difference in incidence of anastomotic leak of 3% for patients with pelvic anastomoses and 2.5% for those with intraabdominal anastomoses [42].

**Proper vascularization of digestive segments** involved in anastomosis is an important factor that can determine healing on the digestive suture. Recently, measurement of microcirculation has gained substantial interest. Vignali et al. measured transmural colonic blood flow by a laser-Doppler flowmetry technique before bowel manipulation and after vascular ligation and transection [90]. They observed a significant difference (*p* < 0.001) in mean rectal stump flow reduction after colonic division of 16% in patients who developed anastomotic leak compared with 6.2% in patients without anastomotic dehiscence. Other authors have measured microvessel density with immunohistochemical analysis of CD31 expression in the proximal segment of anastomosis, but they did not find any significant correlation with leakage [91]. An experimental study assessing this issue has shown that total microvascular density should not be measured, but rather functional microvascular density [92]. Recent studies have suggested that near-infrared (NIR) imaging using indocyanine green has a potential benefit in the evaluation of vascular perfusion at the anastomotic site [25,93,94]. In a study of 400 patients with colorectal cancer using indocyanine green to assess vascularization of anastomotic margins, the authors showed that 11 patients (2.8%) needed to change the transection line by NIR imaging due to fluorescence abnormalities. The rate of AL was 1%. They have concluded that NIR imaging using indocyanine green may contribute to the reduction of anastomotic leak [95].

**Surgeon’s experience** in colorectal surgery has been claimed by some authors to be a risk factor for anastomotic fistula, with a high-volume colorectal surgeons having a smaller incidence of postoperative complications than low-volume ones [96,97]. Other authors found no statistical difference in anastomotic leak rate between consultants, trainee surgeons and independent surgeons [98].

**Manual versus mechanical** execution of anastomosis is a subject of debate regarding the best results on post-operative anastomotic fistula. Several studies assessing stapled and handsewn colorectal anastomoses found no difference in terms of postoperative leakage rate between the two techniques [99,100]. In a systematic review, Choy et al. compared stapler versus handsewn technique and side-to-side versus end-to-end types of sutures in ileocolic anastomosis [101]. The authors concluded that stapled, functional end-to-end anastomosis is associated with fewer leaks than handsewn ileocolic anastomosis. Puleo et al. analyzed the type of anastomosis technique used in ileocolic anastomosis and found that a handsewn technique was associated with an increased anastomotic leak rate compared with stapled technique [102]. Moreover, they showed that stapled end-to-side configuration was associated with a lower incidence of leak than side-to-side anastomosis. A more recent prospective multicenter international study including 3208 patients evaluated the relationship between leak and anastomosis technique following right-sided colonic resection [103]. The authors found that stapled anastomosis was associated with an increased anastomotic leak rate. Additionally, some authors have reported an increased risk of anastomotic leakage in stapled anastomosis using multiple firing [104,105]. Other authors have shown that there were no significant differences in the rate of anastomotic leakage when comparing either the number or the length of the cartridges used to transect the rectum [106].

**Laparoscopic colorectal** surgery has recently become popular, and many surgeons currently use this approach in colorectal pathology. Despite the fact that laparoscopic surgery for rectal cancer has technical difficulties such as pelvic approach (especially in men), lack of tactile sense or inadequate cutting angle after transection, the benefits are now widely accepted. Randomized controlled trials confirming equivalent oncological outcome and long-term survival between open and laparoscopic surgery have been published [107,108]. Laparoscopy has distinct differences from open surgery, such as the need for multiple stapler firings when transecting the rectum, which is associated with an increased anastomotic leak rate, although this is likely to be reduced with advances in stapler technology [109].

A recent meta-analysis concluded that laparoscopic rectal resection was associated with decreased blood loss, smaller incisions and longer operative times compared with the open approach. No differences were observed for postoperative morbidity and mortality between the two techniques [110]. The COLOR II trial showed statistically significant differences in terms of blood loss, bowel recovery and the length of hospital stay in favor of laparoscopic approach and no difference between open and laparoscopic rectal resection in terms of postoperative anastomotic leakage or mortality [111]. Multiple studies concluded the same things, namely that there were no significant differences between open and laparoscopic rectal resection in terms of anastomotic leakage, postoperative morbidity and mortality [107,112,113]. Two recent meta-analysis comparing laparoscopic intersphincteric resection versus open approach for low rectal cancer have shown no significant difference for anastomotic leakage incidence between the two groups [114,115].

Modern techniques such as transanal-TME and robotics are receiving worldwide attention recently and may represent an alternative to laparoscopy, especially if they are proven to be oncologically safe, clinically advantageous for the patient and less challenging for the surgeon [116,117,118]. Robotic colorectal surgery is safe and feasible but has no clear advantages compared with laparoscopic surgery in terms of postoperative outcomes and complications [119,120]. Studies showed that the rate of anastomotic leakage was comparable between the two techniques [121,122].

Anastomotic dehiscence remains one of the most significant complications after low anterior resection (LAR) for rectal cancer. Making (constructing) a **protective stoma** (ileostomy or colostomy) after LAR remains a subject of debate. Even experienced surgeons find it difficult to predict which patients will develop an anastomotic leak, but studies have demonstrated that low anastomosis has a considerably higher risk of dehiscence [123,124,125]. There are studies and meta-analyses that showed a decreased anastomotic leak rate when surgeons used a defunctioning stoma in LAR by diverting the fecal stream and reducing the intraluminal pressure of the bowel [16,99,126,127]. In a multicenter prospective study including rectal cancer patients with anastomoses below 8 cm, leak rate was 5.8% in the stoma group and 16.3% in the no stoma group. Leakage rates and reoperation rates for leakage were significantly higher in the group without a stoma. With multivariate analysis, the authors found that male gender and the absence of a stoma were significantly associated with anastomotic leakage [128].

The most commonly used type of stoma is the defunctioning loop ileostomy. Several meta-analyses have compared ileostomies with colostomies and concluded that ileostomy is preferred in terms of reduced stoma-related morbidity [129,130]. Some publications have reported that the overall leakage and reoperation rates were similar in patients with or without a protective stoma [131,132]. Moreover, diverting stoma construction and closure is associated with increased morbidity and cost [133,134]. The potential disadvantages of a protective stoma include the need for another operation, a longer hospital stay and stoma-related complications, such as prolapse, stenosis, peristomal abscess, parastomal hernia and skin problems. Although it does not reduce the risk of anastomotic fistula, the diverting stoma diminishes its clinical consequences [135,136,137,138].

The role of **prophylactic pelvic drainage** in reducing the postoperative complication rate after rectal surgery remains controversial. New strategies in rectal cancer management including total mesorectal excision (TME) and neoadjuvant chemoradiotherapy have led to a higher rate of sphincter-saving procedures [139]. The utilization of a prophylactic drain reduces extraperitoneal fluid collections, limiting the risk of consequent contamination [140,141]. On the other hand, in the case of anastomotic failure, drainage might help in its early detection and thereby facilitate its proper and early management [140,141]. The role of pelvic drainage in reducing the incidence of infraperitoneal anastomotic leakage and pelvic sepsis has been sustained by some authors [99,106,141,142]. A recent meta-analysis of randomized controlled trials comparing drainage versus no drainage after rectal surgery found an anastomotic leakage rate of 14.8% in the drain group and 16.7% in the no-drain group (*p* = 0.37). The authors concluded that prophylactic use of pelvic drainage after extraperitoneal colorectal anastomosis has no impact on the incidence of anastomotic leak [143]. A meta-analysis by Guerra et al. suggests that pelvic drainage does not confer any significant advantage in the prevention of postoperative complications and may even add to the postoperative morbidity of patients receiving rectal surgery with extraperitoneal anastomoses [144].

According to a study by Denost et al., the overall interval between surgery and the diagnosis of postoperative pelvic sepsis was 7.8 ± 5.4 days for drained patients and 6.7 ± 3.3 days for undrained patients, and the average delay to reintervention was shorter for patients without pelvic drains [145]. Although this difference had no statistical significance, it suggests a trend to delayed diagnosis of anastomotic leak in patients with pelvic drainage. This prospective randomized trial failed to demonstrate the superiority of the pelvic drainage after low anterior resection for rectal cancer. The authors recommend not using pelvic drain after rectal excision for cancer, except in the case of operative bleeding or beyond TME surgery [145].

**Emergency surgery** in case of peritonitis and/or bowel obstruction places patients at a higher risk of adverse postoperative events. Emergency resection was shown to be an independent risk factor for anastomotic failure in some studies [16,42] and, moreover, an independent risk factor for death after leakage [16]. In a prospective study on 1417 patients with colorectal cancer, Choi et al. found that emergency surgery and a high ASA grade of 3 to 5 are independent factors associated with an increased incidence of leakage. They concluded that a temporary diverting stoma to protect the primary anastomosis or even avoidance of anastomosis could be considered for patients with the two risk factors present [146]. Anastomosis is not necessarily contraindicated in emergency circumstances. There are multiple studies on the feasibility of anastomosis with a defunctioning stoma for peritonitis due to perforated diverticulitis [147,148]. However, when performing an emergency resection, the surgeon should evaluate what patients are at a high risk for leakage, and in this situation, use of a temporary defunctioning stoma and avoidance of an anastomosis are sensible and safe options.

**Operative time** longer than 3 h has also been described in the literature as being associated with an increased incidence of anastomotic dehiscence [42,48,59,149,150]. Midura et al. categorized anastomotic leaks as minor and major and found that open approach and operative time more than 3 h were associated with both types of leaks [59].

## 4. Postoperative Risk Factors

**Anemia** has been described as a risk factor for leaks. Hemoglobin is related to perfusion and oxygenation of the anastomotic margins, an essential factor for anastomotic healing. Currently, this is a subject of assessment, and several authors have shown that a hemoglobin level less than 11 g/dL increases risk of leak, as explained by a decreased capacity to transport oxygen to the tissues and subsequent risk of ischemia [151,152].

Operative blood loss and **blood transfusions** were both independently associated with an increased risk of anastomotic failure [28,153]. Blood loss may induce ischemia at the anastomoses and hence impaired anastomotic healing. Blood transfusions may induce immunological suppression, thereby increasing the risk of infectious conditions around anastomoses [31,154]. In their study, Jannasch et al. found a 1.5-fold risk for anastomotic leak in patients with blood transfusion, without differentiation of the amount of blood units given [21].

## 5. Conclusions

Anastomotic leak after colorectal surgery, a major complication with increased postoperative morbidity and mortality still remains a challenge despite surgical progress and technological advances. The awareness of risk factors should influence treatment and procedure-related decisions. Further research is required to focus on risk factors that currently are insufficiently explored, to reduce the risk and subsequent effects associated with anastomotic leakage.

## Figures and Tables

**Table 1 diagnostics-11-02382-t001:** Risk factors associated with increased risk of postoperative anastomotic leakage (AL) after colorectal surgery.

Risk Factors for AL	Pre-Operative	Intra-Operative	Post-Operative
Modifiable	Smoking Alcohol consumption Obesity Malnutrition Seric albumin and protein level NSAIDs Mechanical bowel preparation Preoperative chemoradiotherapy	Vascularization of digestive segments Type of suture (manual/mechanical) Type of approach (open/laparoscopic) Prophylactic pelvic drainage Diverting stoma Blood transfusions	Anemia Blood transfusions
Non-modifiable	Male gender ASA score > II Charlson Comorbidity Index History of radiotherapy	Distance of tumor from the anal verge Emergency/elective surgery Operative time Surgeon experience	

**Table 2 diagnostics-11-02382-t002:** Comparative results concerning the impact of mechanical bowel preparation on anastomotic leakage.

Authors, Year	Type of Study	Location of Anastomosis	No of Patients	No Prep. AL (%)	MBP+/ABX- AL, *n* (%)	MBP+/ABX+ AL, *n* (%)	Adjusted OR (95% CI)	*p*
Kiran RP et al., 2015 [64]	Retrospective	Colorectal	8442	2296 4.6%	3822 3.5%	2424 49 (2.1%)	0.57 (0.35–0.94)	**0.026**
Ji WB et al., 2017 [72]	Retrospective	Rectal	1369	831 9.3%	538 7.8%	–		0.349
Klinger AL et al., 2019 [74]	Retrospective	Colorectal	27,804	5471	7617	8855	0.53(0.43–0.65)	**<0.001**
Garfimkle R et al., 2017 [75]	Retrospective	Colorectal	40,446	13,219 4.4%	13,935 3.7%	11,720 2.3%	0.53 (0.44–0.63)	**<0.001**
Toh JW et al., 2018 [76]	Retrospective	Colorectal	5729	1295	1713	2721	Laparoscopic: 0.42 (0.19–0.94) Open: 0.30 (0.12–0.77)	**0.035** **0.012**
Scarborough JE et al., 2015 [65]	Prospective	Colorectal	4999	1092 5.7%	2322 4.2%	1494 2.8%	0.48 (0.32–0.73)	**0.001**
Rollins KE et al., 2018 [70]	Meta-analysis	Colorectal	21,568	7793 4.8%	2475	11,300 3.7%	0.90 (0.74–1.10)	0.32

ABX—antibiotic treatment; AL—anastomotic leakage; MBP—mechanical bowel preparation.

**Table 3 diagnostics-11-02382-t003:** List of clinical trials comparing the impact of neoadjuvant therapy on anastomotic leakage.

Authors, Year	Type of Study	No of Patients	pR(C)T+Surgery AL %	Surgery Alone AL %	OR, 95% CI	*p*	pR(C)T Increase AL
Marijnen CA et al., 2002 [81]	Prospective randomized trial	1414	695 11%	719 12%	-	-	NS
Chang JS et al., 2014 [82]	Retrospective	1437	360 7.5%	1077 5.9%	-	-	NS
Qin Q et al., 2016 [80]	Randomized controlled trial	318	201 13%	117 4.2%	OR = 3.50 (95% CI, 1.20–10.19)	**0.02**	Yes
Park EJ et al., 2018 [79]	Retrospective	2035	427 13.2%	1608 6.3%	OR = 1.84 (95% CI, 1.26–2.69)	**0.002**	Yes
Qin C et al., 2014 [86]	Meta-analysis	3375	1660 8.6%	1715 8.4%	OR = 1.02 (95% CI, 0.80–1.30)	0.88	NS
Hu MH et al., 2017 [83]	Meta-analysis	9675	3743 10.6%	5932 8.54%	OR = 1.16 (95% CI, 0.99–1.36)	0.07	NS

NS—not significant; pR(C)T—preoperative radio(chemo)therapy.

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
