# Peer review of "Updates of Risk Factors for Anastomotic Leakage after Colorectal Surgery"

_diagnostics, 2021, doi:10.3390/diagnostics11122382_

Round 1

Reviewer 1 Report

Advantage:

  1. Nearly all risk factors about anastomotic leakage showed in the past articles are all collected
  2. Positive or negative risk factors are mentioned in the article
  3. Enough references are enrolled
  4. This article is suitable as a systemic review for young surgeons

Disadvantage:

  1. The preoperative chemoradiotherapy includes two divisions: long course CCRT and short CCRT. The authors did not show them clearly. It is important in recent trend in CRC treatment (such as total chemotherapy before operation in advanced tumor) and long course CCRT should have high anastomotic leakage rate.
  2. The author should give the readers proper recommendation in each topic ( such as mechanical bowel preparation may be used or may not in colon or rectal surgery)

Author Response

Dear Madam/Sir,

Thank you very much for your comments. We have performed a significant English language review. As suggested, we have included a discussion regarding the MBP and the long/short course CCRT. Once again, thank you for your valuable observations that further improved the paper.

Respectfully,

Narcis Zarnescu

Reviewer 2 Report

To the Authors

The reviewer thinks that this paper is compact with a good review of previous papers.

So, this article meets the value of “Diagnostics”

Author Response

Dear Madam/Sir,

Thank you very much for your comments. We have performed a significant English language review.

Respectfully,

Narcis Zarnescu

Reviewer 3 Report

I would like to congratulate the authors on the difficult task of reviewing the risk factors for anastomotic leakage. However, this is a narratvie review, and it lacks the methods on the selection of relevant literature. As such, it may create significant bias in reporting some results and not reporting others and thus mislead the authors and readers unintentionally. I would recommend re-doing the study as a systematic review, clearly stating, what were the search terms, what were the inclusion and exclusion criteria for the cited studies and then proceed with the review.

Author Response

Dear Madam/Sir,

Thank you very much for your comments. We have performed a significant English language review.

I am totally agreeing with you that a systematic review is significantly better than a narrative review. However, the methodology is so different, that makes this pursue to end up into a different type of article. By reviewing the literature, one can identify several systematic reviews on subtopics of risk factors associated with anastomotic fistula. I list here few examples published in 2018 only.

Leenen JPL, Hentzen JEKR, Ockhuijsen HDL. Effectiveness of mechanical bowel preparation versus no preparation on anastomotic leakage in colorectal surgery: a systematic review and meta-analysis. Updates Surg. 2018 Mar 21. doi: 10.1007/s13304-018-0526-4.

Modasi A, Pace D, Godwin M, Smith C, Curtis B. NSAID administration postcolorectal surgery increases anastomotic leak rate: systematic review/meta-analysis. Surg Endosc. 2018 Jul 11. doi: 10.1007/s00464-018-6355-1.

Slesser AA, Pellino G, Shariq O, Cocker D, Kontovounisios C, Rasheed S, Tekkis PP. Compression versus hand-sewn and stapled anastomosis in colorectal surgery: a systematic review and meta-analysis of randomized controlled trials. Tech Coloproctol. 2016 Oct;20(10):667-76. doi: 10.1007/s10151-016-1521-8. Epub 2016 Aug 23.

McSorley ST, Steele CW, McMahon AJ., Meta-analysis of oral antibiotics, in combination with preoperative intravenous antibiotics and mechanical bowel preparation the day before surgery, compared with intravenous antibiotics and mechanical bowel preparation alone to reduce surgical-site infections in elective colorectal surgery. BJS Open. 2018 May 10;2(4):185-194. doi: 10.1002/bjs5.68. eCollection 2018 Aug. Review.

 Any attempt to write a systematic review on all risk factors associated with anastomotic fistula would transform this material in a very long book chapter due to the methodology required by design. Once again, thank you for your valuable observations.

Respectfully,

Narcis Zarnescu

Reviewer 4 Report

Authors presented comperhensive review regarding the AL in colorectal surgery. The paper widely discuss increasing end decreasing risk factors for AL.

IMHO the summering figure or table of factors related with AL and their significances will improve the paper.

Also the paragraph with proposition of the best clinical approach to decrease the AL in colorectal cancer should be added.

Author Response

Dear Madam/Sir,

Thank you very much for your comments. We have performed a significant English language review. As suggested, we have added a summering table of factors related with AL. We have added several paragraphs regarding the clinical approach (MBP, preoperative radiotherapy). Once again, thank you for your valuable observations that further improved the paper.

Respectfully,

Narcis Zarnescu